

# Persistent and extreme outliers in causes of death by state, 1999–2013

Francis P. Boscoe

New York State Cancer Registry, New York State Department of Health, Albany, NY, United States

## ABSTRACT

In the United States, state-specific mortality rates that are high relative to national rates can result from legitimate reasons or from variability in coding practices. This paper identifies instances of state-specific mortality rates that were at least twice the national rate in each of three consecutive five-year periods (termed persistent outliers), along with rates that were at least five times the national rate in at least one five-year period (termed extreme outliers). The resulting set of 71 outliers, 12 of which appear on both lists, illuminates mortality variations within the country, including some that are amenable to improvement either because they represent preventable causes of death or highlight weaknesses in coding techniques. Because the approach used here is based on relative rather than absolute mortality, it is not dominated by the most common causes of death such as heart disease and cancer.

## INTRODUCTION

This paper builds upon the findings of the paper, "The Most Distinctive Causes of Death by State, 2001–2010," published in the online journal *Preventing Chronic Disease* in May, 2015 (*Boscoe & Pradhan, 2015*). That paper—formally a "GIS Snapshot," consisting of a single map and accompanying short description—presented the most distinctive cause of death for each state and the District of Columbia for the 2001–2010 period. "Most distinctive" was defined as the highest ratio of state-specific death rate to national death rate for each of the causes of death included in the 113 Selected Causes of Death List published by the *National Center for Health Statistics (2002)*. For example, the age-adjusted death rate due to pneumoconiosis nationwide was 0.3 per 100,000, but in Kentucky it was 1.0 and in West Virginia it was 3.9. The respective ratios of 3.3 and 12.4 were higher than for any other cause of death in these states, making them the most distinctive. The mapped causes of death can also be understood as those with the highest state-specific relative risks, the highest location quotients (*Mayer & Pleeter, 1975*), or as the largest outliers. In general, the identification of outliers is useful for assessing the integrity of a data set and to identify genuinely unusual phenomena that can give rise to hypotheses (*Osborne & Overbay, 2004*).

In the time since the original paper was first submitted for publication, three additional years of data have become available. I incorporate these data into an alternative way of conducting the analysis that identifies what I term persistent outliers and extreme outliers.

Corresponding author
Francis P. Boscoe,
francis.boscoe@health.ny.gov

Persistent outliers were those causes of death with an age-adjusted rate that was at least twice the national rate in each of the five-year time periods 1999–2003, 2004–2008, and 2009–2013. Extreme outliers were defined as those causes of death with an age-adjusted rate that was at least 5 times above the national rate in at least one of the time periods. Identifying all of the outliers in this manner instead of identifying exactly one per state, as was done on the original map, is a more inclusive means of summarizing the data.

## METHODS

National and state-specific age-adjusted death rates for all of the causes of death included in the 113 Selected Causes of Death List for the period 1999–2013 were obtained from the *Centers for Disease Control and Prevention* (CDC) Wide-ranging Online Data for Epidemiologic Research (WONDER) web site (CDC, no date). This list was developed for the general analysis of mortality data and for ranking causes of death and is based on International Classification of Diseases version 10 (ICD-10) codes. The list includes some overlapping cause of death categories; including these results in a total of 136 causes of death. Data were divided into three 5-year periods: 1999–2003, 2004–2008, and 2009–2013. The ratios of the state rates to the national rates for each cause of death in each period were calculated, and persistent and extreme outliers were identified, as defined above. State-level counts below 10 were suppressed by WONDER and excluded from the analysis; fewer than 0.01% of deaths fell into this category, which included causes of death so rare that no data was reported for any state (as with measles, 12 deaths nationally in 15 years) or only reported for a few of the largest states (as with whooping cough, with data only reported for California and Texas). State-level counts between 11 and 19, marked by WONDER as "unreliable," were included in the analysis. 95% confidence intervals around the ratios were determined using the RELRISK option in the FREQ procedure in SAS version 9.3 (SAS Institute, Cary, North Carolina, USA). Results were tabulated for all causes of death in the list, even where the classifications overlapped, as for example with homicide, homicide by firearm, and homicide by other and unspecified means. An exception was made for "other and unspecified events of undetermined intent and their sequelae" and "events of undetermined intent," because these two categories were nearly identical—the first comprised over 99% of the second. Only the second, more inclusive category is reported here.

## RESULTS

There were 62 persistent outliers among 28 states plus the District of Columbia (Table 1). The District of Columbia had the most persistent outliers, with 9, while there were 22 states without any. There were 38 extreme outliers among 14 states plus the District of Columbia (Table 2). The District of Columbia led with 7, while 36 states did not have any. Twelve of the persistent outliers also appeared on the list of extreme outliers: water and air accidents (Alaska), events of undetermined intent (Maryland and Utah), other acute ischemic heart disease (Oklahoma and Virginia), influenza (South Dakota), and pneumoconiosis (West Virginia), plus five in the District of Columbia—HIV, homicide, homicide by firearm, hypertensive heart disease, and atherosclerotic cardiovascular disease. Table 2 also reveals

Boscoe (2015), *PeerJ*, DOI 10.7717/peerj.1336

**Table 1** Persistent outliers, by state, 1999–2013.

| State | Cause of death | 1999–2003 | | 2004–2008 | | 2009–2013 | |
| --- | --- | --- | --- | --- | --- | --- | --- |
| | | Deaths | RR | Deaths | RR | Deaths | RR |
| Alabama | Symptoms, signs and abnormal clinical and laboratory findings, not elsewhere classified (R00–R99) | 5,550 | 2.31 (2.25–2.37) | 6,829 | 2.69 (2.63–2.76) | 7,882 | 2.88 (2.81–2.94) |
| Alabama | Accidental discharge of firearms (W32–W34) | 209 | 3.81 (3.31–4.38) | 161 | 3.27 (2.79–3.83) | 127 | 2.47 (2.07–2.95) |
| Alabama | Other heart diseases (I26–I51) | 28,255 | 2.11 (2.09–2.14) | 30,196 | 2.28 (2.26–2.31) | 30,702 | 2.12 (2.10–2.14) |
| Alabama | Heart failure (I50) | 10,894 | 2.44 (2.39–2.48) | 10,225 | 2.31 (2.27–2.36) | 9,981 | 2.05 (2.01–2.09) |
| Alaska | Water, air and space, and other and unspecified transport accidents and their sequelae (V90–V99, Y85) | 186 | 9.19 (7.95–10.63) | 124 | 6.09 (5.10–7.27) | 129 | 6.09 (5.12–7.25) |
| Alaska | Tuberculosis (A16–A19) | 13 | 2.81 (1.63–4.85) | 13 | 4.17 (2.42–7.19) | 18 | 3.70 (2.33–5.88) |
| Alaska | Accidental drowning and submersion (W65–W74) | 119 | 3.11 (2.60–3.73) | 134 | 3.44 (2.90–4.08) | 118 | 2.95 (2.46–3.54) |
| Alaska | Intentional self-harm (suicide) by discharge of firearms (X72–X74) | 387 | 2.16 (1.96–2.39) | 470 | 2.53 (2.31–2.77) | 503 | 2.49 (2.28–2.72) |
| Alaska | Other and unspecified nontransport accidents and their sequelae (W20–W31, W35–W64, W75–W99, X10–X39, X50–X59, Y86) | 301 | 2.18 (1.95–2.44) | 292 | 2.04 (1.82–2.29) | 320 | 2.07 (1.86–2.31) |
| Arizona | Discharge of firearms, undetermined intent (Y22–Y24) | 62 | 2.52 (1.95–3.25) | 80 | 3.02 (2.41–3.79) | 63 | 2.24 (1.74–2.88) |
| Arkansas | Discharge of firearms, undetermined intent (Y22–Y24) | 37 | 3.10 (2.24–4.30) | 40 | 3.46 (2.53–4.74) | 27 | 2.04 (1.40–2.99) |
| District of Columbia | Hypertensive heart and renal disease (I13) | 144 | 4.82 (4.09–5.68) | 140 | 5.16 (4.37–6.09) | 131 | 4.47 (3.76–5.31) |
| District of Columbia | Human immunodeficiency virus (HIV) disease (B20–B24) | 1,221 | 8.49 (8.02–8.99) | 1,015 | 8.97 (8.43–9.55) | 536 | 4.41 (4.05–4.81) |
| District of Columbia | Atherosclerotic cardiovascular disease, so described (I25.0) | 2,794 | 4.06 (3.91–4.21) | 2,797 | 5.06 (4.88–5.25) | 2,338 | 3.96 (3.80–4.13) |
| District of Columbia | Hypertensive heart disease (I11) | 1,375 | 5.44 (5.16–5.73) | 936 | 3.38 (3.17–3.60) | 852 | 2.87 (2.69–3.07) |
| District of Columbia | Assault (homicide) by discharge of firearms (U01.4, X93–X95) | 756 | 5.92 (5.51–6.36) | 650 | 4.75 (4.39–5.13) | 381 | 2.50 (2.26–2.77) |
| District of Columbia | Assault (homicide) (U01–U02, X85–Y09, Y87.1) | 990 | 5.01 (4.71–5.34) | 827 | 4.24 (3.96–4.54) | 535 | 2.49 (2.29–2.72) |
| District of Columbia | Assault (homicide) by other and unspecified means and their sequelae (U01.0–U01.3, U01.5–U01.9, U02, X85–X92, X96–Y09, Y87.1) | 234 | 3.49 (3.07–3.97) | 177 | 3.16 (2.72–3.66) | 154 | 2.48 (2.12–2.91) |

Boscoe (2015), *PeerJ*, DOI 10.7717/peerj.1336

| State | Cause of death | 1999–2003 | | 2004–2008 | | 2009–2013 | |
|---|---|---|---|---|---|---|---|
| | | Deaths | RR | Deaths | RR | Deaths | RR |
| District of Columbia | Viral hepatitis (B15–B19) | 111 | 2.09 (1.74–2.52) | 157 | 2.63 (2.24–3.07) | 168 | 2.47 (2.12–2.87) |
| District of Columbia | Pregnancy, childbirth and the puerperium (O00–O99) | 10 | 2.52 (1.35–4.69) | 28 | 3.65 (2.51–5.29) | 21 | 2.37 (1.54–3.63) |
| Hawaii | Accidental drowning and submersion (W65–W74) | 153 | 2.06 (1.75–2.41) | 169 | 2.18 (1.87–2.54) | 227 | 2.77 (2.43–3.16) |
| Idaho | Water, air and space, and other and unspecified transport accidents and their sequelae (V90–V99, Y85) | 99 | 2.35 (1.93–2.86) | 113 | 2.63 (2.19–3.17) | 115 | 2.41 (2.01–2.90) |
| Iowa | Influenza (J09–J11) | 196 | 2.46 (2.13–2.83) | 178 | 2.39 (2.06–2.78) | 190 | 2.59 (2.24–2.99) |
| Kansas | Atherosclerosis (I70) | 1,706 | 2.16 (2.06–2.27) | 1,700 | 3.32 (3.16–3.48) | 1,941 | 3.56 (3.40–3.73) |
| Kentucky | Pneumoconioses and chemical effects (J60–J66, J68) | 240 | 3.01 (2.65–3.43) | 216 | 3.27 (2.86–3.75) | 211 | 2.88 (2.51–3.31) |
| Louisiana | Accidental discharge of firearms (W32–W34) | 179 | 3.15 (2.72–3.66) | 195 | 3.99 (3.45–4.61) | 169 | 3.41 (2.92–3.98) |
| Louisiana | Assault (homicide) by discharge of firearms (U01.4, X93–X95) | 2,109 | 2.35 (2.25–2.45) | 2,298 | 2.47 (2.37–2.58) | 2,186 | 2.34 (2.24–2.44) |
| Louisiana | Meningococcal infection (A39) | 27 | 2.91 (1.99–4.27) | 21 | 3.51 (2.27–5.43) | 10 | 2.05 (1.10–3.83) |
| Louisiana | Assault (homicide) (U01–U02, X85–Y09, Y87.1) | 2,834 | 2.00 (1.93–2.08) | 2,916 | 2.14 (2.07–2.22) | 2,765 | 2.03 (1.95–2.10) |
| Maine | Influenza (J09–J11) | 67 | 2.15 (1.69–2.73) | 105 | 3.50 (2.89–4.25) | 90 | 2.72 (2.21–3.35) |
| Maryland | Events of undetermined intent (Y10–Y34, Y87.2, Y89.9) | 3,144 | 7.37 (7.10–7.65) | 3,405 | 6.99 (6.74–7.24) | 2,990 | 5.88 (5.66–6.11) |
| Minnesota | Influenza (J09–J11) | 293 | 2.57 (2.28–2.89) | 281 | 2.62 (2.33–2.95) | 272 | 2.41 (2.14–2.73) |
| Mississippi | Discharge of firearms, undetermined intent (Y22–Y24) | 37 | 2.77 (2.00–3.84) | 34 | 2.53 (1.80–3.55) | 34 | 2.73 (1.94–3.84) |
| Mississippi | Hypertensive heart and renal disease (I13) | 317 | 2.18 (1.95–2.43) | 324 | 2.37 (2.12–2.64) | 378 | 2.55 (2.31–2.83) |
| Mississippi | Accidental discharge of firearms (W32–W34) | 133 | 3.75 (3.15–4.45) | 108 | 3.30 (2.72–4.00) | 83 | 2.53 (2.03–3.14) |
| Mississippi | Accidental exposure to smoke, fire and flames (X00–X09) | 452 | 2.80 (2.55–3.07) | 398 | 2.66 (2.41–2.94) | 377 | 2.39 (2.16–2.65) |
| Mississippi | Hypertensive heart disease (I11) | 2,624 | 2.16 (2.08–2.25) | 3,416 | 2.46 (2.38–2.55) | 3,349 | 2.24 (2.17–2.32) |
| Mississippi | Heart failure (I50) | 6,566 | 2.42 (2.36–2.28) | 6,879 | 2.62 (2.56–2.68) | 6,339 | 2.24 (2.18–2.29) |
| Montana | Influenza (J09–J11) | 72 | 3.47 (2.75–4.38) | 43 | 2.08 (1.54–2.80) | 72 | 3.21 (2.54–4.05) |

Boscoe (2015), *PeerJ*, DOI 10.7717/peerj.1336

| State | Cause of death | 1999–2003 | | 2004–2008 | | 2009–2013 | |
|---|---|---|---|---|---|---|---|
| | | Deaths | RR | Deaths | RR | Deaths | RR |
| Montana | Intentional self-harm (suicide) by discharge of firearms (X72–X74) | 571 | 2.11 (1.94–2.29) | 594 | 2.14 (1.97–2.32) | 729 | 2.44 (2.26–2.62) |
| Montana | Water, air and space, and other and unspecified transport accidents and their sequelae (V90–V99, Y85) | 64 | 2.07 (1.62–2.65) | 79 | 2.64 (2.12–3.30) | 76 | 2.43 (1.94–3.05) |
| Montana | Accidental discharge of firearms (W32–W34) | 25 | 2.18 (1.47–3.23) | 22 | 2.02 (1.33–3.08) | 24 | 2.34 (1.57–3.50) |
| Nebraska | Influenza (J09–J11) | 115 | 2.72 (2.26–3.27) | 86 | 2.10 (1.70–2.60) | 96 | 2.31 (1.89–2.83) |
| Nebraska | Symptoms, signs and abnormal clinical and laboratory findings, not elsewhere classified (R00–R99) | 2,391 | 2.33 (2.23–2.42) | 2,228 | 2.05 (1.97–2.14) | 2,398 | 2.03 (1.95–2.11) |
| Nevada | Legal intervention (Y35, Y89.0) | 35 | 2.46 (1.76–3.44) | 42 | 2.72 (2.01–3.69) | 38 | 2.36 (1.71–3.25) |
| New Mexico | Legal intervention (Y35, Y89.0) | 36 | 2.99 (2.15–4.15) | 34 | 2.84 (2.02–3.98) | 50 | 4.33 (3.27–5.73) |
| New Mexico | Alcoholic liver disease (K70) | 932 | 2.41 (2.26–2.57) | 969 | 2.24 (2.10–2.39) | 1,275 | 2.72 (2.57–2.87) |
| New Mexico | Accidental poisoning and exposure to noxious substances (X40–X49) | 1,268 | 2.65 (2.51–2.80) | 1,823 | 2.16 (2.06–2.26) | 2,261 | 2.56 (2.45–2.67) |
| North Dakota | Influenza (J09–J11) | 42 | 2.37 (1.75–3.21) | 47 | 2.95 (2.21–3.93) | 38 | 2.34 (1.70–3.22) |
| Oklahoma | Other acute ischemic heart diseases (I24) | 5,324 | 25.20 (24.44–25.99) | 4,130 | 18.52 (17.90–19.15) | 2,156 | 8.95 (8.56–9.35) |
| Oregon | Meningococcal infection (A39) | 16 | 2.37 (1.45–3.88) | 12 | 3.12 (1.76–5.52) | 12 | 2.80 (1.58–4.96) |
| South Carolina | Other acute ischemic heart diseases (I24) | 796 | 3.43 (3.20–3.69) | 998 | 3.67 (3.44–3.91) | 1,356 | 4.37 (4.13–4.62) |
| South Dakota | Influenza (J09–J11) | 102 | 5.07 (4.17–6.17) | 78 | 4.18 (3.35–5.23) | 91 | 4.57 (3.72–5.63) |
| Utah | Events of undetermined intent (Y10–Y34, Y87.2, Y89.9) | 889 | 5.65 (5.28–6.04) | 1,472 | 7.44 (7.06–7.84) | 875 | 4.00 (3.74–4.28) |
| Utah | Symptoms, signs and abnormal clinical and laboratory findings, not elsewhere classified (R00–R99) | 2,275 | 2.60 (2.49–2.71) | 3,645 | 3.72 (3.60-3.84) | 3,427 | 2.97 (2.87–3.07) |
| Vermont | Hyperplasia of prostate (N40) | 12 | 2.78 (1.57–4.90) | 15 | 2.94 (1.77–4.89) | 19 | 3.24 (2.06–5.08) |
| Vermont | Influenza (J09–J11) | 38 | 2.80 (2.03–3.85) | 36 | 2.86 (2.06–3.97) | 33 | 2.28 (1.62–3.21) |
| Virginia | Other acute ischemic heart diseases (I24) | 2,419 | 6.21 (5.95–6.48) | 2,699 | 6.14 (5.89–6.39) | 2,592 | 5.19 (4.98–5.41) |

Boscoe (2015), *PeerJ*, DOI 10.7717/peerj.1336

Table 1 (*continued*)

| State | Cause of death | 1999–2003 | | 2004–2008 | | 2009–2013 | |
|---|---|---|---|---|---|---|---|
| | | Deaths | RR | Deaths | RR | Deaths | RR |
| Virginia | Pneumoconioses and chemical effects (J60–J66, J68) | 332 | 2.62 (2.35–2.93) | 271 | 2.51 (2.22–2.84) | 286 | 2.31 (2.05–2.60) |
| West Virginia | Pneumoconioses and chemical effects (J60–J66, J68) | 557 | 12.90 (11.82–14.07) | 414 | 12.08 (10.93–13.35) | 338 | 9.36 (8.38–10.45) |
| Wyoming | Influenza (J09–J11) | 34 | 3.45 (2.46–4.83) | 19 | 2.13 (1.36–3.34) | 38 | 3.52 (2.56–4.84) |
| Wyoming | Intentional self-harm (suicide) by discharge of firearms (X72-X74) | 331 | 2.24 (2.01–2.49) | 330 | 2.19 (1.97–2.44) | 440 | 2.65 (2.42–2.91) |

Boscoe (2015), *PeerJ*, DOI 10.7717/peerj.1336

**Table 2  Extreme outliers, by state, 1999–2013.**

| State | Cause of Death | 1999–2003 | | 2004–2008 | | 2009–2013 | |
|---|---|---|---|---|---|---|---|
| | | Deaths | RR | Deaths | RR | Deaths | RR |
| Alaska | Water, air and space, and other and unspecified transport accidents and their sequelae (V90–V99, Y85) | 186 | **9.19 (7.95–10.63)** | 124 | **6.09 (5.10–7.27)** | 129 | **6.09 (5.12–7.25)** |
| Alaska | Discharge of firearms, undetermined intent (Y22–Y24) | 19 | **6.68 (4.25–10.51)** | <10 | | 27 | **8.30 (5.67–12.15)** |
| District of Columbia | Human immunodeficiency virus (HIV) disease (B20–B24) | 1,221 | **8.49 (8.02–8.99)** | 1,015 | **8.97 (8.43–9.55)** | 536 | 4.41 (4.05–4.80) |
| District of Columbia | Assault (homicide) by discharge of firearms (U01.4, X93–X95) | 756 | **5.92 (5.51–6.36)** | 650 | 4.24 (3.96–4.54) | 381 | 2.50 (2.26–2.77) |
| District of Columbia | Assault (homicide) (U01–U02, X85–Y09, Y87.1) | 990 | **5.01 (4.71–5.34)** | 827 | 4.74 (4.39–5.13) | 535 | 2.49 (2.29–2.72) |
| District of Columbia | Hypertensive heart disease (I11) | 1,375 | **5.44 (5.16–5.73)** | 140 | **5.16 (4.37–6.09)** | 852 | 2.87 (2.69–3.07) |
| District of Columbia | Atherosclerotic cardiovascular disease, so described (I25.0) | 2,794 | 4.06 (3.91–4.21) | 2,797 | 5.06 (4.88–5.25) | 2,338 | 3.96 (3.80–4.13) |
| Iowa | Other and unspecified acute lower respiratory infections (J22, U04) | 17 | 2.76 (1.70–4.46) | 16 | **5.32 (3.23–8.77)** | <10 | |
| Kansas | Other and unspecified acute lower respiratory infections (J22, U04) | 58 | **10.87 (8.30–14.23)** | 17 | **6.09 (3.74–9.89)** | <10 | |
| Louisiana | Syphilis (A50–A53) | 13 | **9.23 (5.26–16.19)** | <10 | | 16 | **9.29 (5.58–15.45)** |
| Maryland | Events of undetermined intent (Y10–Y34, Y87.2, Y89.9) | 3,144 | **7.37 (7.10–7.65)** | 3,405 | **6.99 (6.74–7.24)** | 2,990 | **5.88 (5.66–6.11)** |
| Maryland | Syphilis (A50–A53) | <10 | | 10 | **5.65 (2.99–10.65)** | <10 | |
| Massachusetts | Events of undetermined intent (Y10–Y34, Y87.2, Y89.9) | 2,762 | **5.54 (5.33–5.77)** | 950 | 1.71 (1.61–1.83) | 429 | 0.75 |
| Nebraska | Other and unspecified acute lower respiratory infections (J22, U04) | 24 | **6.72 (4.47–10.11)** | 20 | **9.92 (6.33–15.54)** | <10 | |
| Oklahoma | Other acute ischemic heart diseases (I24) | 5,324 | **25.20 (24.44–25.99)** | 4,130 | **18.52 (17.90–19.15)** | 2,156 | **8.95 (8.56–9.35)** |
| Rhode Island | Events of undetermined intent (Y10–Y34, Y87.2, Y89.9) | 425 | **5.24 (4.76–5.77)** | 271 | 3.01 (2.67–3.40) | 60 | 0.65 |
| South Dakota | Influenza (J09–J11) | 102 | **5.07 (4.17–6.17)** | 78 | 4.18 (3.35–5.23) | 91 | 4.57 (3.72–5.62) |
| Utah | Events of undetermined intent (Y10–Y34, Y87.2, Y89.9) | 889 | **5.65 (5.28–6.04)** | 1,472 | **7.44 (7.06–7.84)** | 875 | 4.00 (3.74–4.28) |
| Vermont | Other nutritional deficiencies (E50–E64) | 11 | 4.77 (2.63–8.63) | <10 | | 27 | **12.59 (8.59–18.46)** |
| Virginia | Other acute ischemic heart diseases (I24) | 2,419 | **6.21 (5.95–6.48)** | 2,699 | **6.14 (5.89–6.39)** | 2,592 | **5.19 (4.98–5.41)** |
| West Virginia | Pneumoconioses and chemical effects (J60–J66, J68) | 557 | **12.90 (11.82–14.07)** | 414 | **12.08 (10.93–13.35)** | 338 | **9.36 (8.38–10.45)** |

that the number of extreme outliers has decreased over time. Between 1999 and 2003, there were 17; from 2004–2008 there were 13; and from 2009–2013 there were 8.

## DISCUSSION

The tables highlight instances where state mortality rates exceeded national rates by substantial margins. These can be understood as either genuine phenomena—where the risk of death due to a certain cause was truly elevated—or as artifacts of state-specific coding practices. The former category includes unambiguous infectious and chronic diseases such as viral hepatitis and pneumoconiosis, and well-specified types of accidents such as accidental drowning and exposure to smoke, fire and flames. The latter category includes causes of death containing the words "other," "unspecified" and "unknown," where a state, for whatever reason, was unable to code deaths to the same level of specificity as other states.

There are a number of possible explanations for this lack of specificity. The information could have truly been absent—a physician or coroner might have only indicated something like "cardiac arrest" on the death certificate, for example, and there were insufficient resources to follow up and obtain something more precise. It is also possible that coding guidelines may have been interpreted overly strictly or literally, or may have been perceived as unclear, outcomes that are influenced by the experience level of the death certifier (*Johnson et al., 2010*). There could have also been instances of "motivated misreporting," in which the person filling out the death certificate may have had an incentive to be vague (*Osborne & Overbay, 2004*). An example of this has occurred in Maryland, where the state's chief medical examiner is on record that many "events of undetermined intent"—which include unresolved homicides, suicides, and accidents—cannot be coded more specifically without input from the legal system, even though the medical determination of intent is distinct from the legal determination (*Fenton, 2012*). Critics have argued that this practice substantially suppresses the official homicide rate. Indeed, Maryland's rate of "events of undetermined intent" was 6–7 times above the national average in each of the 3 time periods.

For some of the reported outliers, it is not obvious whether the findings were genuine, an artifact, or some combination of the two. For example, influenza, which appeared as an outlier in 9 different states (Iowa, Maine, Minnesota, Montana, Nebraska, North Dakota, South Dakota, Vermont, and Wyoming), would seem to be a clearly defined cause of death. Yet the number of deaths due to influenza is small, totaling 3,697 in 2013 (*Centers for Disease Control and Prevention, 2015*). Influenza deaths are perceived as common because people tend to be more familiar with the counts of influenza and pneumonia combined (56,979 in 2013, placing it among the top ten causes of death nationwide when so grouped), or the number of influenza-associated deaths (estimated at 20,000–30,000 annually, a number derived from mathematical models rather than death certificates *Doshi, 2008*; *Thompson et al., 2009*; *Centers for Disease Control and Prevention, 2015*). Of the comparatively small number of deaths officially ascribed to influenza, a minority were confirmed with a lab test (these receive ICD-10 codes J09 and J10), while the remainder

were based on observation (these receive code J11). The nine states with unusually high influenza death rates may simply have been more aggressive in ordering lab tests, or more willing to have called influenza-like illness influenza, than to have had a true excess risk.

Note that this analysis was only able to identify likely examples of substantial overreporting in certain causes of death. There have also been well-documented examples of substantial underreporting, such as with suicide (*Klugman, Condran & Wray, 2013*), pregnancy-related deaths (*Deneux-Tharaux et al., 2005*), and injuries from falls (*Betz, Kelly & Fisher, 2008*). In some cases, such as with "events of undetermined intent," the overreported category can imply which categories were likely underreported, but a separate analysis would be required to identify properly these negative outliers; such an analysis would be complicated by the suppression of counts less than 10. For the present analysis, the suppression of counts less than 10 might have masked some potentially interesting information (for example, if the 12 measles deaths had been concentrated in just a few states), but by definition would not have included anything of widespread public health importance.

Each one of the causes of death highlighted in the tables suggests a story about mortality disparities, mortality coding disparities, or some combination of the two that demands further investigation. In the interest of brevity, I will comment only on the dozen entries which appeared in both tables. The District of Columbia, with 5 of the 12, revealed itself as an outlier among outliers. Although not a state, its data are typically reported with the 50 states, as was done here. It is unique among "states" in having an African–American majority and being entirely urban. It also has the highest poverty rate and income inequality of any "state," making it an outlier by numerous measures. The high rates of HIV-related deaths and homicide seen here reflect the urban pathologies of intravenous drug use and crime, while hypertensive heart disease and atherosclerotic cardiovascular disease reflect DC's racial composition, even while the precise reasons for greater hypertension among black Americans remain elusive (*Fuchs, 2011*).

Moving to Alaska, the classification "water, air and space, and other and unspecified transport accidents and their sequelae" has a straightforward explanation: travel by water and air is vastly more common here than in other states, and is the only way to reach many settlements within the state. Pneumoconioses, more commonly known as black lung disease, has a similarly obvious association with West Virginia, the state most closely associated with coal mining. "Events of undetermined intent," with high rates in Maryland and Utah, has already been discussed, as has influenza, with particularly high rates in South Dakota.

That leaves "other acute ischemic heart disease," which appeared in both Oklahoma and Virginia. From 1995 to 1999, the rate for this cause of death was over 25 times the national average in Oklahoma, making it the most extreme outlier in this entire analysis. The rate subsequently dropped to 9 times the national average in 2008–2013, still one of the more extreme values. This is a clear example of coding imprecision, reflecting an inability to distinguish among chronic heart disease, heart attack (myocardial infarction), and a few other less common conditions in a manner not shared by other states. For any studies

which distinguish among types of ischemic heart disease (see, for example, *Ibfelt, Bonde & Hansen, 2010*), care would have to be taken to make sure that the results were not biased by the data from these two states.

The need for uniform standards for cause of death coding, and for public health data generally, is obvious—in order to compare conditions in different places and times, the measurement of those conditions must be done in as similar a manner as possible. The drop in the number of extreme outliers over time suggests that standardization has been improving. Massachusetts and Rhode Island represent good examples of this trend. Both of these states had very high rates of "events of undetermined intent" in 1999–2003, but by 2009–2013 were below the national average. Public health agencies are continuously trying to improve standards and data quality; for example, since 2012, the National Center for Health Statistics has been flagging rare causes of death such as those caused by vaccine preventable diseases and requesting that states follow up and attempt to verify them (*Centers for Disease Control and Prevention, 2014*). Findings such as those reported here can also serve to motivate improvements, as no state wants to be identified as an outlier for a preventable cause of death or an indicator of low data quality.

Cause of death coding based on the International Classification of Diseases is the most widely used system in the world and enables comparisons between countries and across decades. While no such system can ever be perfect or tell us everything we would like to know, it is in our collective interest to strive for the highest data quality possible.

### Funding
This project was supported in part by CDC's National Program of Cancer Registries through cooperative agreement number 5U58DP003879 awarded to the New York State Department of Health. The funders had no role in study design, data collection and analysis, decision to publish, or preparation of the manuscript.

### Grant Disclosures
The following grant information was disclosed by the author:
CDC's National Program of Cancer Registries: 5U58DP003879.

### Competing Interests
The author declares there are no competing interests.

### Author Contributions
- Francis P. Boscoe conceived and designed the experiments, performed the experiments, analyzed the data, contributed reagents/materials/analysis tools, wrote the paper, prepared figures and/or tables, reviewed drafts of the paper.

### Data Availability
CDCWonder: http://wonder.cdc.gov/ucd-icd10.html

The contents of Table 1: http://figshare.com/articles/Persistent_outliers_among_state_level_causes_of_death_1999_2013/1411231

The contents of Table 2: http://figshare.com/articles/Extreme_outliers_among_state_level_causes_of_death_1999_2013/1422036.

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
