# Peer review of "Persistent and extreme outliers in causes of death by state, 1999–2013"

_PeerJ, doi:10.7717/peerj.1336_

## Round 0.1 · original submission · Minor Revisions

Thank you for your submission which has been well received by our reviewers. We would like you to address the points raised by reviewer 1 regarding data suppression and reliability. Reviewer 2 suggests that you consider removing the figures, however I think that these do add to the understanding of the data (particularly to readers with limited knowledge of US geography), therefore I will leave it up to you to decide whether to adopt their suggested revision.

Reviewer 1 ·

Basic reporting

The article appears to adhere to all PeerJ policies listed. It is clearly written and makes a convincing argument about the problems that arise from inconsistencies in how deaths are coded across the country.

Experimental design

Research question is clearly defined. Methods described with sufficient detail.

Some additional comments that may help the reader understand the data a little better:

1. It will be helpful if the author provided brief comments on the impact of suppression. For example, how many data points were removed from the analysis and would they have any significant impact on the analysis.

2. When using data from CDC WONDER, rates are sometimes marked as "unreliable" - did the author encounter any instances of such rates?

Validity of the findings

No additional comments other than the ones noted in the "Experimental Design" section.

·

Basic reporting

This article more or less adheres to PeerJ policies regarding basic reporting. However, I would suggest to the editor and the author that the figures are redundant, since the same information is presented in tables, and this is one of the rare instances where tables are more informative than graphs. What is the point of showing a map of the United States, putting a symbol for various causes of death in states with persistent or extreme outliers, and force the reviewer to consult a lengthy legend at the bottom?

Experimental design

This article does not involve an experimental design. It is an analysis of publicly available mortality rates for all of the United States plus Washington, D.C. However, in my discipline (sociology) this investigation would still be considered "primary research".

The analyses are fairly basic. It classifies states as being persistent or extreme outliers based on the ratio of the state's age-adjusted death rate to the nationwide age-adjusted death rate (for various causes of death).

This kind of presentation is straight-forward and has the virtue of being understandable by lay audiences (it also has the virtue of being easily reproducible; I was able to reproduce one of the relative risks presented in the paper). I don't see how the findings could be artifacts of the analysis.

Validity of the findings

The data is fine, and Boscoe is thoughtful when discussing why some states have persistent and/or extreme outliers, acknowledging that sometimes the rates may reflect truly higher levels of a cause of death (e.g. black lung disease in W Virginia) or they may reflect measurement error.

Additional comments

I thought the findings about the drop in the number of extreme outliers was interesting, suggesting that death statistic collection is becoming increasingly standardized across locales within states.

·

Basic reporting

No comment

Experimental design

No comment

Validity of the findings

No comment

Additional comments

This is a well-written paper with clear, replicable methodologies that identifies “persistent” and “extreme” outliers (defined by the author) of age-adjusted state mortality rates as compared to national rates and explores whether the observed rates might be due to a true excess or to differing coding practices among states.

---

## Round 0.2 · accepted · Accept

Thank you for the revised version of your paper. I am pleased to accept this paper for publication.